# Independent Prototype Propagation
# for Zero-Shot Compositionality

**Frank Ruis**
University of Twente & TNO
research@frank-ruis.nl

**Gertjan J. Burghouts**
TNO
gertjan.burghouts@tno.nl

**Doina Bucur**
University of Twente
d.bucur@utwente.nl

## Abstract

Humans are good at compositional zero-shot reasoning; someone who has never seen a zebra before could nevertheless recognize one when we tell them it looks like a horse with black and white stripes. Machine learning systems, on the other hand, usually leverage spurious correlations in the training data, and while such correlations can help recognize objects in context, they hurt generalization. To be able to deal with underspecified datasets while still leveraging contextual clues during classification, we propose ProtoProp, a novel prototype propagation graph method. First we learn prototypical representations of objects (e.g., zebra) that are independent w.r.t. their attribute labels (e.g., stripes) and vice versa. Next we propagate the independent prototypes through a compositional graph, to learn compositional prototypes of novel attribute-object combinations that reflect the dependencies of the target distribution. The method does not rely on any external data, such as class hierarchy graphs or pretrained word embeddings. We evaluate our approach on AO-Clevr, a synthetic and strongly visual dataset with clean labels, UT-Zappos, a noisy real-world dataset of fine-grained shoe types, and C-GQA, a large-scale object detection dataset modified for compositional zero-shot learning. We show that in the generalized compositional zero-shot setting we outperform state-of-the-art results, and through ablations we show the importance of each part of the method and their contribution to the final results. The code is available on github[1].

## 1 Introduction

As humans, hearing the phrase 'a tiny pink penguin reading a book' can conjure up a vivid image, even though we have likely never seen such a creature before. This is because humans can compose their knowledge of a small number of visual primitives to recognize novel concepts [1], a property which Lake et al. [2] argue is one of the key building blocks for human intelligence missing in current artificial intelligence systems. Machines, on the other hand, are largely data-driven, and usually require many labeled examples from various viewpoints and lighting conditions in order to recognize novel concepts. Since visual concepts follow a long-tailed distribution [3, 4], such an approach makes it near impossible to gather sufficient examples for all possible concepts. Compounded by that, in the absence of sufficient data, vanilla convolutional neural networks will use any correlation they can find to classify training samples, even when they are spurious [5]. In this work, we aim to tackle both of these issues.

Compositional Zero-Shot Learning (CZSL) [6] is the problem of learning to model novel objects and their attributes as a composition of visual primitives. Previous works in CZSL [6, 7, 8] largely ignore the dependencies between classes with shared visual primitives, and the spurious correlations between

---

[1]https://github.com/FrankRuis/protoprop

35th Conference on Neural Information Processing Systems (NeurIPS 2021).

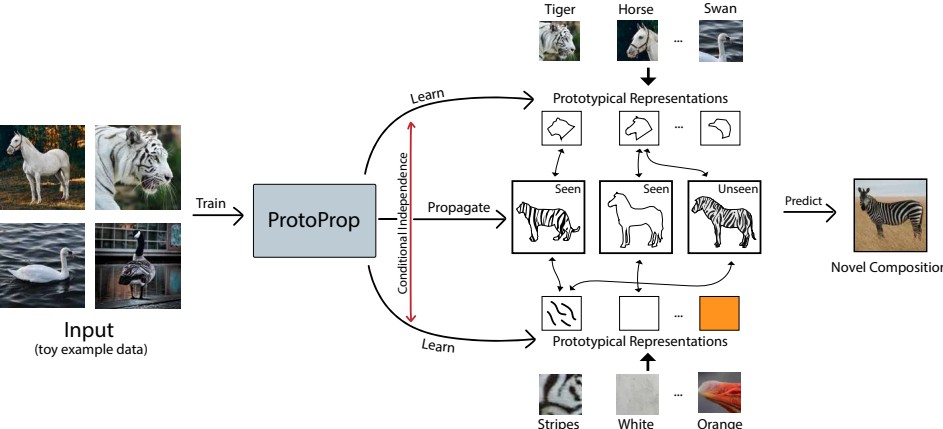

Figure 1: **ProtoProp** A sketch of our proposed method; we learn independent prototypical representations of visual primitives in the form of objects (e.g., horse) and attributes (e.g., stripes). The prototypes are then propagated through a compositional graph, where they are combined into novel compositional prototypes to recognize both seen and unseen classes (e.g., zebra).

attributes and objects. More recently, Atzmon et al. [9] tackle the latter by ensuring conditional independence between attribute and object representations, while Naeem et al. [10] explicitly promote dependencies between the primitives and their compositions. While the independence approach improves generalization, it hurts accuracy on seen classes by removing useful correlations. The explicit dependencies, on the other hand, can share these useful correlations with unseen classes, but there will always be some that are spurious, hurting generalization.

In this work, we propose to take advantage of the strengths of both approaches, respectively, by learning independent visual representations of objects and attributes, and by learning their compositions for the target classes. First, we represent visual primitives by learning local independent prototypical representations. Prototype networks [11] learn an embedding function, where inputs of the same class cluster around one prototypical representation of that class. Here we adopt such a function for learning prototypes of objects and attributes. Next, we leverage a compositional graph to learn the dependencies between the independent prototypes on the one hand, and the desired seen and unseen classes on the other, by propagating the prototypes to compositional nodes. Here, the compositional graph allows some information to be shared between objects that share attributes, e.g., between tigers and zebras. Sylvain et al. [12] show the importance of locality and compositionality for model generalization in zero-shot learning. They also propose a measure for the compositionality of a representation, which is equivalent to our compositional loss function. The proposed method, ProtoProp, is outlined in Figure 1.

**Our main contributions are**: 1) We propose a novel graph propagation method that learns to combine local, independent, attribute and object prototypes into one compositional prototype that can accurately detect unseen compositional classes. 2) A spatial attention-based pooling method that allows us to obtain differentiable attribute and object patches for use in an independence loss function. 3) Our method effectively deals with bias from an underspecified dataset by learning independent representations that then take on the dependencies of the desired target distribution. 4) We validate through ablations the importance of each part of the method (local prototypes vs semantic embeddings, independence loss, backbone finetuning) and their contribution to the final results. 5) We show that we improve on state-of-the-art results on three challenging compositional zero-shot learning benchmarks: 2.5 to 20.2% harmonic mean improvement on AO-Clevr [9], 3.1% harmonic mean improvement on UT-Zappos [13], and a slight improvement on C-GQA compared to the best existing method.

## 2 Related work

**Compositional zero-shot learning** (CZSL) methods aim to recognize unseen compositions from known visual primitives in the form of attributes and objects. One line of work considers embedding the visual primitives in the image feature space. Misra et al. [6] use the weight vectors of linear SVMs as embeddings for the visual primitives, which they transform to recognize unseen compositions. Li et al. [8] look at attribute-object compositions through the lens of symmetry inspired by group theory. A different line of work considers a joint embedding function on the image, attribute, and object triplet, allowing the model to learn dependencies between the image and its visual primitives. Purushwalkam et al. [7] train a set of modular networks together with a gating network that can 'rewire' the classifier conditioned on the input attribute and object pair. Atzmon et al. [9] take a causal view of CZSL, trying to answer which intervention caused the image. They apply conditional independence constraints to the representations of the visual primitives, sacrificing accuracy on the seen data but performing well on unseen data by removing correlations that are useful but hurt generalization to novel compositions. Naeem et al. [10], on the other hand, explicitly promote the dependency between all primitives and their compositions within a graph structure, though they rely on pretrained semantic embeddings which may differ in distribution from the visual concepts they describe.

Our proposed method combines the strengths from Atzmon et al. [9] and Naeem et al. [10] by first learning independent representations of the visual primitives, which are then propagated through a compositional graph to learn the dependency structure of the desired target distribution. Unlike most other methods, we are not reliant on pretrained word embeddings, but instead learn the representations for our visual primitives directly from the training data. Instead of a linear kernel like Atzmon et al. [9], we use a Gaussian kernel for our independence loss as we find that its ability to capture higher-order statistics is beneficial in terms of accuracy. Like Naeem et al. [10], we train our model fully end-to-end, including the feature extractor, as learning a good embedding is often more beneficial than an overly complicated method applied to suboptimal embeddings [14].

**Prototypical networks** [11] aim to learn an embedding function where inputs of the same class cluster around one prototypical representation of that class. They measure L2 distance between samples in pixel space, which is sensitive to non-semantic similarities between images, such as objects from different classes with a similar background and light conditions. Li et al. [15] move the similarity metric to latent space and utilise an autoencoder to directly visualize the prototypes, but they only test on simple MNIST-like benchmarks. Chen et al. [16] extend the method to multiple local prototypes per class, where prototypes are compared to local image patches instead of the entire average-pooled output of a CNN, and evaluate on the more complicated fine-grained bird classification dataset CUB [17]. We take a similar local-prototype approach, but we use direct attribute and object supervision, enabling parameter sharing between classes and allowing the prototypes to be used for zero-shot classification. While for most methods these prototypes are used for the final classification, in our case they are an intermediate representation.

**Graph neural networks** (GNNs) are models that can work directly on the structure of a graph. They have been first proposed by Gori et al. [18], later elaborated upon by Scarselli et al. [19] and popularised by Kipf and Welling [20] in their work on the Graph Convolutional Neural Network (GCN). GNNs grow more popular every year, and a lot of improvements have been proposed recently [21]. Just like the GCNs were inspired by CNNs, most improvements are inspired by other areas of deep learning such as attention mechanisms in Graph Attention Networks [22], but this jump from existing deep learning fields to GNNs has overlooked simpler methods. Wu et al. [23] have taken a step back and stripped down GNNs to their simplest parts by removing nonlinearities and collapsing weight matrices until all that was left was feature propagation followed by a linear model. In the same vein, Huang et al. [24] show that a simple Multilayer Perceptron (MLP) ignoring graph structure followed by a label propagation post-processing step often greatly outperforms GNNs. These simplified methods are mostly limited to transductive settings (test nodes are available at train time) and graphs that exhibit strong homophily (similar nodes are connected). Many real-world graphs fit those criteria, including the graph we use in this work.

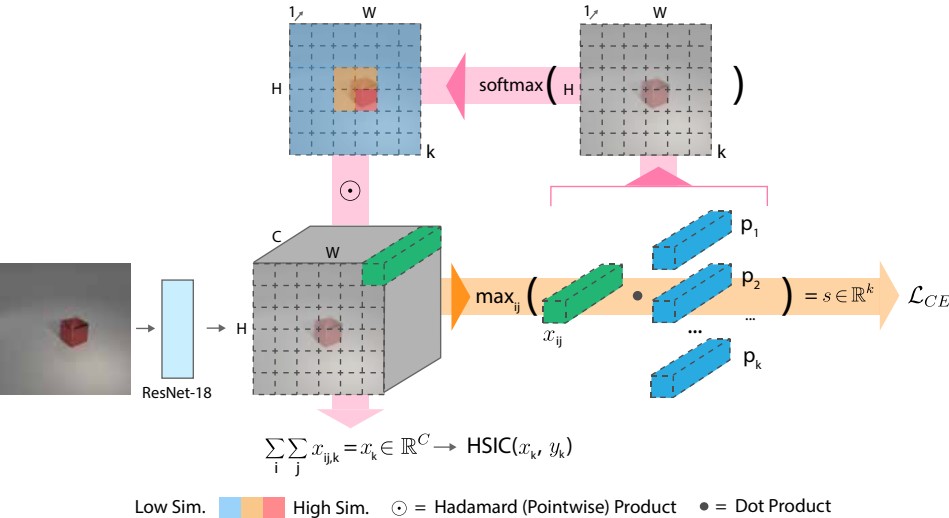

Low Sim. ▮▮▮ High Sim.  ⊙ = Hadamard (Pointwise) Product  • = Dot Product

Figure 2: **Local prototypes with softmax pooling**: Local image patches $x_{ij}$ from the layer just before the average pooling layer of a ResNet-18 are compared to prototype vectors $p_k$ through a similarity function, here a dot product. This outputs a compatibility score $s$, which is optimized via $\mathcal{L}_{CE}$ (cf. Section 3.1). Similar to spatial attention, these patch-prototype compatibility scores are passed through a softmax function and via Hadamard (pointwise) product a weighted sum $z_k$ of the original feature map is calculated, which are passed to the HSIC function to promote independence between object and attribute prototypes (cf. Section 3.2)

## 3 Method

In compositional zero-shot learning (CZSL), we have a set of images $X$ and compositional classes with compositional labels $Y \subseteq A \times O$, where $A$ is a set of attribute labels and $O$ a set of object labels. The labels are subdivided into $Y = Y_s \cup Y_u$, where $Y_s$ are the seen labels for the training set and $Y_u$ unseen labels for the validation and test sets, with $Y_s \cap Y_u = \emptyset$. We denote the training data consisting of only seen compositional classes as $X_s$. Finally, CZSL assumes that each attribute and object is seen in at least one training data point, or more formally $\forall p \in A \cup O \; \exists y \in Y_s$ s.t. $p \cap y \neq \emptyset$.

### 3.1 Prototype-based representations

In this section we describe how we learn local attribute and object prototypes, before they are propagated and combined into compositional prototypes in Section 3.3. Prototypes are an average representation of a target class, with strong generalization and interpretability properties. We employ a local prototype approach similar to Chen et al. [16], though we use direct supervision, encouraging the feature extractor to learn local image representations that encode the attribute and object labels. We use a ResNet-18 [25] backbone for fair comparison to other CZSL methods, but the method is backbone agnostic.

Figure 2 shows an example of a prototype layer. The layer has a set of prototype vectors $\mathbf{P} = \{p_j\}_{j=1}^k$, $p_j \in \mathbb{R}^C$ where each target class is represented by one prototype. $k = |Y_i|$ is the number of attribute or object targets, and $i \in \{0, 1\}$ indicates if we are training on attribute or object labels respectively. The layer takes as input the $H \times W \times C$ output of a CNN just before its final pooling layer, and calculates a compatibility score $s = \langle x_{ij}, p_k \rangle$ (e.g., cosine similarity, L2 norm, dot product) between the input patches $x_{ij} \in \mathbb{R}^C$ over the spatial dimensions $H \times W$ and the prototypes. We find that a dot product similarity metric leads to better generalization. To save on parameters and avoid overfitting, we do not use a fully connected layer to combine prototype similarity scores into a final prediction. Instead, the maximum patch-prototype similarity score is used directly as the compatibility score for the corresponding prototype's class. This compatibility score is optimized through a cross-entropy

loss function:

$$\mathcal{L}_{CE} = \frac{1}{|S|} \sum_{s,y \in S} - \log \left( \frac{\exp(s[y_i])}{\sum_j^{|s|} \exp(s[j])} \right) \tag{1}$$

Here, with $f_p$ as our prototype layer, $S = \{s, y \in (f_p(X_s) \mid Y_s)\}$ is the set of prototype similarity scores and target labels for the training data. $i \in \{0, 1\}$ indicates if we are training on attribute or object labels respectively.

Cross-entropy loss naturally encourages clustering between samples of the same class and separation from other classes, but we find that an additional loss to push these properties even more is beneficial. For that purpose we adopt the cluster and separation costs proposed by Chen et al. [16]:

$$\text{Clst} = \frac{1}{|X|} \sum_{i=1}^{|X|} \min_{j:p_j \in \mathbf{P}_{y_i}} \min_{z \in x_{ij}} \|z - p_j\|_2^2 \qquad \text{Sep} = -\frac{1}{|X|} \sum_{i=1}^{|X|} \min_{j:p_j \notin \mathbf{P}_{y_i}} \min_{z \in x_{ij}} \|z - p_j\|_2^2 \tag{2}$$

The costs Clst and Sep encourage a higher degree of clustering between prototypes of the same class and a greater distance to prototypes of different classes, respectively. The separation cost is only applied to the object prototypes, as the attribute prototypes benefit from less distant representations.

Our Hilbert-Schmidt Independence Criterion (HSIC) loss (details in Section 3.2) requires the input patches with the highest prototype similarity scores as its input, but that would require the use of the argmax function which has a gradient of 0 almost everywhere. Similar to spatial attention, we use the softmax function over the similarity map as a differentiable proxy. The softmax outputs used as weights in a weighted sum of the input patches $x_{ij}$ serve as an approximation of the input patch with the highest similarity score (cf. Figure 2), and ensure that each patch containing information about attribute or object labels is affected by the independence loss proportional to the strength of their appearance (i.e. their similarity score).

### 3.2 Attribute-object independence

To combat the bias in the training data we leverage a differentiable independence metric such that the model can learn representations for the attributes of images that are uniformly distributed w.r.t. their object labels and vice versa.

The Hilbert-Schmidt Independence Criterion (HSIC) [26] is a kernel statistical test of independence between two random variables $A$ and $B$. In the infinite sample limit $\text{HSIC}(A, B) = 0 \iff A \perp\!\!\!\perp B$ as long as the chosen kernel is universal in the sense of Steinwart [27]. We use a Gaussian kernel, as we find that its ability to capture higher-order statistics is beneficial in terms of accuracy, and follow Gretton et al. [26] in setting the kernel size to the median distance between points in a batch. See the Appendix for more details. We define our independence loss function as follows, slightly deviating from Atzmon et al. [9]:

$$\mathcal{L}_{hsic} = \lambda_h \frac{\text{HSIC}(z_a, O) + \text{HSIC}(z_o, A)}{2} \tag{3}$$

where $\lambda_h$ is a hyperparameter controlling the contribution of $\mathcal{L}_{hsic}$ to the final loss, $z_a$ is the softmax-pooled output of the attribute prototype layer (cf. Section 3.1), $z_o$ the softmax-pooled output of the object prototype layer, and $O$ and $A$ the corresponding one-hot encodings of the object and attribute labels respectively.

### 3.3 Prototype propagation graph

Figure 3 shows the full architecture. Two types of prototype layers are trained, one on the object labels and one on the attribute labels, leaving us with centroids with confounding information removed. The prior knowledge about the compositional target classes and the attributes they afford can be represented as a compositional graph $G = (\mathbf{P}_a \cup \mathbf{P}_o \cup \mathbf{C}_y, E)$. It consists of the attribute prototypes $\mathbf{P}_a$, object prototypes $\mathbf{P}_o$, and compositional classes $\mathbf{C}_y \subseteq A \times O$ (a subset of which have no training samples). The edges are undirected, $E = \{x, y \mid x \in \mathbf{P}_a \cup \mathbf{P}_o, y \in \mathbf{C}_y : x \in y\}$, i.e. all attribute (e.g., red) and object (e.g., cube) nodes are connected to the compositional classes they are a part of (e.g., red cube). This corresponds to a bipartite graph with the attribute and object prototypes in one

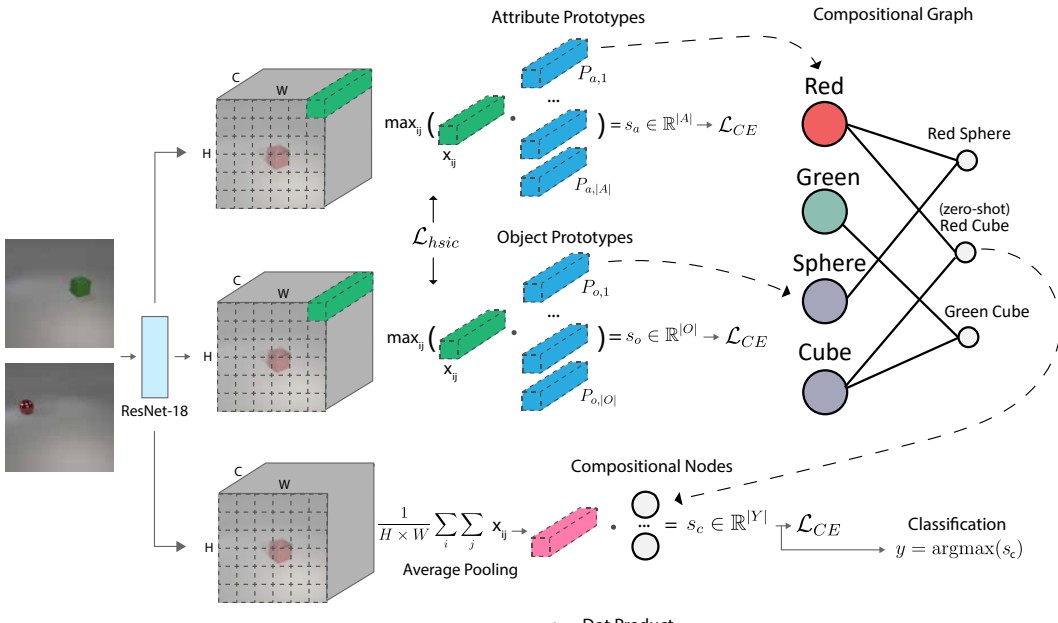

Figure 3: **ProtoProp: overview of proposed method.** Local attribute and object prototypes are trained with independence (HSIC) loss and mapped to nodes in a simplified graph neural network, which learns to combine them into compositional prototypes of both seen and unseen classes. The maximum score in the dot product similarity map between the compositional prototypes and average-pooled backbone output is the final classification prediction.

set and the compositional classes in the other. In the case of AO-Clevr it is a complete bipartite graph, but for real-world datasets such as UT-Zappos this is not the case as, e.g., not all shoes are available in all materials.

As defined above, the independent prototypes are mapped directly to the nodes of the graph (cf. Figure 3), and through shared weights a Graph Neural Network (GNN) learns to propagate the prototypes to the compositional nodes to form new compositional prototypes. The compositional nodes are initialized with zeros. Our GNN is a 2-layer GCN [20]; our findings are in line with the experiments of Naeem et al. [10], where they show that a 2-layer GNN performs best. Inspired by SGC [23] we remove all nonlinearities, as we find that simple linear combinations lead to better generalization:

$$\mathbf{X}' = \hat{\mathbf{D}}^{-1/2}\hat{\mathbf{A}}\hat{\mathbf{D}}^{-1/2}\mathbf{X}\mathbf{\Theta} \tag{4}$$

Here $\mathbf{X}$ are the input nodes, $\mathbf{\Theta}$ a learnable weight matrix, $\hat{\mathbf{A}} = \mathbf{A} + \mathbf{I}$ denotes the adjacency matrix with inserted self-loops, and $\hat{D}_{ii} = \sum_j \hat{A}_{ij}$ its diagonal degree matrix.

We then compute another dot product similarity map $s_c$ between the compositional prototypes and the average-pooled output of the backbone, which is again optimized via cross-entropy loss function. Our final compositional classification is $y = \text{argmax}(s_c)$, i.e., the class corresponding to the compositional prototype with the maximum similarity score. The shared weights in the GNN ensure that the model learns a general composition of attributes and objects that can generalize to unseen compositional classes. By initializing the node features with independent prototypes, the graph is able to learn the dependencies encoded by the compositional graph, including the novel zero-shot classes, instead of the biases of the training data.

# 4 Experiments

## 4.1 Implementation

We implement our model using the PyTorch library [28], using some of the boilerplate code provided by Nagarajan and Grauman [29] and Purushwalkam et al. [7]. The GNN is implemented using PyTorch Geometric [30]. For our independence loss we use a normalized implementation of HSIC provided by Ma et al. [31]. We use the Adam [32] optimizer. Experiments have been run on an 11GB GeForce GTX 1080 Ti graphics card. On AO-Clevr we reach our best result in 1-5 epochs on all splits, at ~5 minutes per epoch. On UT-Zappos we reach our best result in ~30 epochs at ~1.5 minutes per epoch. On C-GQA we reach our best result in 5-10 epochs. See the Appendix for detailed hyperparameters and grid search ranges.

## 4.2 Evaluation

We evaluate our approach on three CZSL datasets. Previous works also evaluate on MIT-States [33], but Atzmon et al. [9] conclude through a large-scale user study that the dataset has a level of ~70% label noise, making it too noisy for evaluating compositionality. All three datasets evaluated in this paper do have some limitations that are relevant for future works, which we describe in the Appendix.

**AO-Clevr** [9, 34] is a synthetic dataset consisting of 3 types of objects (sphere, cube, cylinder) and 8 attributes (red, purple, yellow, blue, green, cyan, gray, brown), with 24 compositional classes in total. There are 6 splits with a varying ratio of unseen to seen classes, ranging from 2:8 to 7:3, allowing insights into the performance of models as the proportion of unseen classes increases.

**UT-Zappos** [13] is a fine-grained dataset of types of shoes with 12 objects (e.g., boat shoes), 16 attributes (e.g., leather), and 116 compositional classes. We use the split proposed by Purushwalkam et al. [7], with 83 seen classes in the training set, 15 seen and 15 unseen classes in the validation set, and 18 seen and 18 unseen classes in the test set.

**C-GQA** [10] is based on GQA [35], a large-scale object detection dataset, that has been modified to fit into the CZSL setting by filtering on bounding boxes of objects that are only described by a single attribute.

**Metrics**: like other recent works we adopt the generalized zero-shot evaluation protocol proposed by Purushwalkam et al. [7], Chao et al. [36]. Instead of evaluating only on the unseen classes as in the Closed setting, the Generalized setting considers both seen and unseen classes in the validation and test sets. To account for the inherent bias towards seen classes, Chao et al. [36] add a calibration bias term to the activations of unseen classes. As the value of the bias is varied between $-\infty$ and $+\infty$, they draw a curve with the accuracy on the unseen classes on the y axis and the accuracy on the seen classes on the x axis, and report the Area Under the Curve (AUC). The harmonic mean is defined as $2 \cdot \frac{Acc_s \cdot Acc_u}{Acc_s + Acc_u}$, where $Acc_s$ is the accuracy on the seen classes and $Acc_u$ the accuracy on the unseen classes. It penalizes large differences between the two metrics and as such indicates how well the model performs on both seen and unseen classes simultaneously. We also follow prior works in reporting the Closed seen accuracy where only the seen classes are considered (corresponding to a bias of $-\infty$) and closed unseen accuracy where only the unseen classes are considered (corresponding to a bias of $+\infty$). For the results on AO-Clevr we report the seen and unseen accuracy components of the harmonic mean in accordance to Atzmon et al. [9]. Atzmon et al. [9] do not perform post-hoc bias calibration, but instead handle the seen-unseen bias during training.

Like Naeem et al. [10], we train our feature extractor end-to-end with the rest of our model. Other methods keep the backbone fixed, but Naeem et al. [10] have shown that they perform worse when allowed to finetune as they will then start to overfit. While our method is designed to take full advantage of the backbone, we show our results with a fixed backbone in our ablations.

## 4.3 Results

For each of the results we select the best model by the best harmonic mean on the validation set and report the results for all metrics on the test set. The results with error bars come from 5 training runs with random initializations. We note that, unlike other methods, Causal [9] does not perform post-hoc

Table 1: **UT-Zappos and C-GQA results**, ProtoProp improves on state-of-the-art results in the AUC and harmonic mean metrics.

| | UT-Zappos | | | | C-GQA | | | |
|---|---|---|---|---|---|---|---|---|
| Method | AUC | Seen | Unseen | Harmonic | AUC | Seen | Unseen | Harmonic |
| AttOp [29] | 25.9 | 59.8 | 54.2 | 40.8 | 0.9 | 11.8 | 3.9 | 2.9 |
| LE+ [6] | 25.7 | 53.0 | 61.9 | 41.0 | 1.2 | 16.1 | 5.0 | 5.3 |
| SymNet [8] | 23.9 | 53.3 | 57.9 | 39.2 | 3.3 | 25.2 | 9.2 | 9.8 |
| TMN [7] | $24.7 \pm 4.4$ | $58.8 \pm 1.4$ | $49.4 \pm 4.7$ | $40.7 \pm 2.0$ | 2.2 | 21.6 | 6.3 | 7.7 |
| Causal [9] | $23.3 \pm 0.3$ | - | $55.4 \pm 0.8$ | $31.8 \pm 1.7$ | — | — | — | — |
| CGE* [10] | $32.5 \pm 1.5$ | $61.0 \pm 0.9$ | $\mathbf{65.9 \pm 1.2}$ | $47.1 \pm 1.7$ | 3.6 | **31.4** | 14.0 | 14.5 |
| ProtoProp* (ours) | $\mathbf{34.7 \pm 0.8}$ | $\mathbf{62.1 \pm 0.9}$ | $65.5 \pm 0.2$ | $\mathbf{50.2 \pm 1.3}$ | $\mathbf{3.7 \pm 0.1}$ | $26.4 \pm 1.2$ | $\mathbf{18.1 \pm 0.9}$ | $\mathbf{15.1 \pm 0.2}$ |

*\* Using finetuned features*

bias calibration, but instead tackles the bias against seen classes during training, which is a benefit of their approach.

**AO-Clevr** Figure 4 shows the results on AO-Clevr. We also report the results for CGE [10] (after a grid search using their own codebase) as they have not reported experiments on this dataset. See the Appendix for the exact values and error bars. We observe that for the 3:7, 5:5, 6:4 and 7:3 splits, one or more colors are not part of the training data, making a large portion of the validation and test classes impossible to classify through compositional methods on those splits. Our method performs better than the compared methods on all splits, with minimal impact on the seen accuracy. The improvements are in the range of 2.5 to 20.2% for the harmonic mean and 3.3 to 17.0% for the unseen accuracy, with higher gains as the proportion of unseen classes increases significantly.

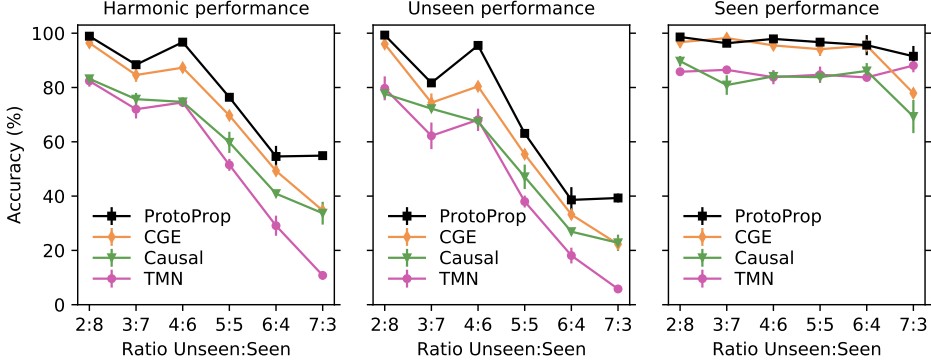

Figure 4: **AO-Clevr results** Plots of the seen and unseen accuracy and their harmonic mean (y-axis) as the ratio of unseen:seen compositional classes (x-axis) increases. ProtoProp consistently outperforms state-of-the-art methods, especially when the portion of unseen classes grows.

**UT-Zappos** Table 1 shows the results on UT-Zappos. Because earlier works only report a single best run, we report the average over 5 random initializations including standard error for the two best performing previous models (CGE and TMN), using their own codebase and reported hyperparameters. We find that, especially for this dataset, the error bars are important since it is highly susceptible to random initialization, where a 'lucky' initialization can result in a significantly larger harmonic mean. Our method outperforms earlier methods on all metrics except for the closed unseen accuracy where CGE performs slightly better. Our method performs especially well when both seen and unseen classes are taken into account, as evidenced by the AUC and harmonic mean improvements.

**C-GQA** Table 1 shows the results on C-GQA. We report the average of our results over 5 random intializations. Because of the scale of the dataset we were not able to do the same for related work, as we do not have enough compute to run experiments using their preferred hyperparameters. Our method trades some seen accuracy for more accuracy on the unseen data, and slightly outperforms the previous best method. We find that the C-GQA dataset has some limitations that makes it less suitable for object-centric classifiers or a single-label setting such as CZSL, which we discuss in the Appendix.

Table 2: **Ablation on the AO-Clevr 4:6 split**: Checkmarks indicate whether prototypes are used as node features, semantic vectors as node features, local prototypes are trained with independence loss, or the backbone is frozen respectively. Prototypes trained when semantic node features are used are not part of the final classification but still improve the feature extractor output.

| Node Features | Indep. Proto | Finetune | Seen | Unseen | Harmonic |
|---|---|---|---|---|---|
| Visual | | ✓ | $94.5 \pm 0.1$ | $77.0 \pm 2.8$ | $84.8 \pm 1.7$ |
| Visual | ✓ | | $78.2 \pm 1.1$ | $73.0 \pm 1.0$ | $75.5 \pm 0.8$ |
| Visual | ✓ | ✓ | $\mathbf{97.9 \pm 1.0}$ | $\mathbf{95.5 \pm 0.9}$ | $\mathbf{96.7 \pm 0.7}$ |
| Semantic | | ✓ | $95.4 \pm 1.1$ | $82.4 \pm 0.7$ | $88.4 \pm 0.1$ |
| Semantic | ✓ | ✓ | $97.3 \pm 1.2$ | $84.5 \pm 1.4$ | $90.4 \pm 0.9$ |

Table 3: **Ablation on the AO-Clevr 4:6 split**: Additional ablations on the linear vs. gaussian kernel for $\mathcal{L}_{hsic}$, dot vs. cosine distance function, and linear vs. nonlinear GNN.

| Ablation | Seen | Unseen | Harmonic |
|---|---|---|---|
| Linear kernel | $97.0 \pm 0.9$ | $71.6 \pm 1.2$ | $82.4 \pm 1.4$ |
| Cosine | $89.2 \pm 3.4$ | $64.0 \pm 2.7$ | $74.5 \pm 3.8$ |
| Nonlinear GNN | $95.0 \pm 0.7$ | $76.2 \pm 0.4$ | $84.6 \pm 0.8$ |
| ProtoProp | $\mathbf{97.9 \pm 1.0}$ | $\mathbf{95.5 \pm 0.9}$ | $\mathbf{96.7 \pm 0.7}$ |

## 4.4 Ablations

To determine the effect of our visual features in the form of prototypes, in contrast to the semantic features in most prior works, we perform ablations where we replace our prototypes with pretrained word embeddings. We also take a look at the importance of the independence loss function and finetuning the backbone.

**Early ablations** Table 3 shows the results of our ablations for several design choices that were mentioned in the method section, specifically the linear vs. gaussian kernel for $\mathcal{L}_{HSIC}$, dot vs. cosine distance function, and linear vs. nonlinear GNN, with all other parameters equal to the final ProtoProp architecture. We note that these choices have been made early on after empirically choosing the option that performed best. The cosine distance metric, for example, was more difficult to optimize and takes twice as long to converge compared to the dot product. Similarly, a nonlinear GNN may be easier to optimize for our method because the node features are prototypes which (as opposed to semantic features) are already in the visual domain, and therefore don't require nonlinear transformations. As such, there may be more optimal settings where these alternative options perform better than we have reported here.

**Importance of prototypes** Table 2 shows the results of our ablations on the 4:6 split on AO-Clevr. Row 1 shows the results for our proposed method without the independence loss function, in row 2 we add the independence loss but freeze the backbone, and row 3 shows the results with both independence and finetuning, as described in the method section. The prototypes perform worse than semantic embeddings when used as node features without the independence loss, but with the independence loss they receive significant gains and come out on top with an $8.3\%$ increase in harmonic mean accuracy over the semantic features. With a frozen backbone the harmonic mean for our method is just slightly higher $(0.8\%)$ than Causal on the same split, though it reaches that accuracy in a fraction of the time.

**Visual vs. semantic** For the ablation in row 4 we use semantic word embeddings (word2vec [37]) as node features, which is equivalent to Naeem et al. [10] with a simplified GCN. In row 5 we also train our local prototypes but don't use them during classification, only to influence the local features the feature extractor learns. Training the local prototypes improves the results slightly even when they are not used for the final classification, by encouraging the backbone to learn local features that represent the target attributes and objects.

**Effect of independence** Table 2 shows that the prediction accuracy breaks down when not using the independence loss. Figure 5 provides some intuition for this through a t-SNE [38] plot of softmax-pooled attribute (in this case color) representations from our model on AO-Clevr. Without the loss there are three strongly separated clusters per color (one per shape), and many colors are embedded closer to different colors of the same shape. When we do use the independence loss, each of the

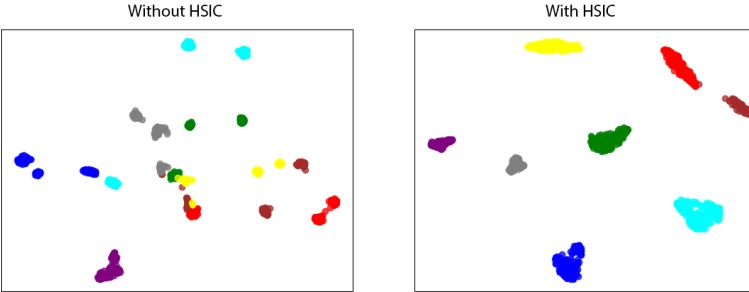

Figure 5: t-SNE plot of softmax-pooled color patches for AO-Clevr that were trained with (left) and without (right) HSIC loss, where with the HSIC loss we get a homogeneous grouping of the attributes and no fragmentation by spurious confounders.

colors is part of their own single cluster and the only correlations left are those between visually similar colors, improving the accuracy of the compositional prototypes after the propagation step.

# 5 Limitations

Like typical CZSL works, our method is limited to a single attribute and object label per image, and each individual attribute and object needs to be seen in at least one training data point. Existing attribute datasets that we have evaluated are often either noisy or do not work well in the aforementioned single-attribute setting. But if this initial hurdle of quality datasets is overcome, compositional methods make it easier to handle the addition of new classes, especially for rare objects with little to no available images. To extend the scope to more realistic settings and other datasets, in future work we would extend the method to multiple attributes, e.g., through attribute prototype regression [39], or by leveraging textual descriptions that can be scraped from the web in a semi-supervised fashion [40].

The hyperparameters can be difficult to tune, as there are hyperparameters for the backbone, prototype layer, independence loss, and compositional GNN. We do find, however, that the prototype layers can be tuned separately from the rest, as the best performing individual prototypes and the hyperparameters that led there also perform best when used in conjunction with the GNN. As such, most hyperparameters can be adopted from existing prototype-based classification works that have been trained on a similar dataset.

# 6 Conclusion

In order to have a chance at recognizing the long-tailed distribution of visual concepts, our models need to become better at recognition through shared visual primitives. To this end, we have proposed a novel prototype propagation method for compositional zero-shot learning. Our method learns prototypes of visual primitives that are independent from the other visual primitives they appear with, and propagates those prototypes through a compositional graph in order to recognize unseen compositions. The method works with just the attribute and object annotations, and, as such, is not reliant on external sources of information, such as hierarchy graphs or pretrained semantic embeddings. We evaluate our work on two CZSL benchmarks, and improve on state-of-the-art results, especially with large fractions of unseen classes, with minimal impact on the accuracy of the seen classes.

**Broader impact**    Although not intended or designed for particular applications that are harmful, our method could be misused, if deployed as part of unethical applications in the real world; this can only be prevented by an ethical review for each application. Furthermore, while we attempt to deal with bias in recognition, this is not a silver bullet, and as such in each application continuous analysis is needed to ensure that all assumptions continue to hold.

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
