# A  Appendix

## A.1  Hyperparameters

**AO-Clevr** Like Atzmon et al. [1] we performed grid searches over the following splits: {2:8, 5:5, 6:4, 7:3}. We used the largest batch size that could fit in memory on our limited hardware, which was 256 for an image size of 224x224. For the learning rate (Adam [2] optimizer) we searched in the range of {0.001, 0.0001, 1e04, 5e-4, 5e-5}, with weight decay {0, 5e-4. 5e-5}. We chose a weight decay of 5e-5 and learning rate of 5e-4 until the 4:6 split and 1e-4 afterwards.

Prototype dimension: {256, 300, 512}, backbone output dimension: {256, 300, 512}, Graph layers: {1, 2, 3}, graph hidden dimension: {256, 512, 1024, 2048, 4096}, $\lambda_h$: {0, 1, 5, 10, 25, 50}, Clst: {0, 0.1, 0.05, 0.01, 0.005}, Sep: {0, 0.1, 0.05, 0.01, 0.005}.

We chose a prototype dimension of 256, backbone output of 512, 2 graph layers, graph hidden dimension of 512, $\lambda_h$ of 10, Clst and Sep of 0.01.

**UT-Zappos** we again used the Adam optimizer, with learning rate in the ranges {5e-5, 5e-4, 5e-3}, and weight decay {0, 5e-4. 5e-5}, where we chose a learning rate and and weight decay of 5e-5 and a batch size of 128. For the rest of the parameters we searched the same ranges as above, where the same choices were optimal as for AO-Clevr.

## A.2  Hilbert-Schmidt Independence Criterion

The (biased) empirical HSIC estimator [3] is defined as:

$$HSIC(U,V) = \frac{1}{m^2} trace(\mathbf{KHLH})$$

Where $\mathbf{K}$ and $\mathbf{L}$ are $m \times m$ matrices with entries $k_{ij}$ and $l_{ij}$, $\mathbf{H} = \mathbf{I}\frac{1}{m}\mathbf{1}\mathbf{1}^\top$ , and $\mathbf{1}$ is a $1 \times m$ vector of ones. The elements of $\mathbf{K}$ and $\mathbf{L}$ are outputs of a kernel function over the inputs $U$ and $V$ such as the (universal) gaussian kernel $k_{ij} := \exp\left(-\sigma^{-2}\|u_i - u_j\|^2\right)$ where $\sigma$ is the kernel size. We follow Gretton et al. [3] in setting the kernel size to the median distance between points, but universality of the gaussian kernel holds for any kernel size. The empirical estimator has a bias in the order of $O(m^{-1})$ which is negligible at even moderate sample sizes.

## A.3  Dataset Limitations

In this section we detail some observations we made for common CZSL datasets which may be useful for future works using these same datasets.

### A.3.1  AO-Clevr

Some of the attributes are not part of the training data for several unseen:seen splits, violating an important CZSL assumption, and making it impossible to recognize certain compositions. Specifically, for the 'unseen:seen-n_seed' seeds, the following fractions of unseen test samples are impossible to compose due to missing attributes:

- $\frac{900}{2400}$ for 3:7-0 and 3:7-1.
- $\frac{900}{3600}$ for 5:5-0 and 5:5-2.
- $\frac{2700}{4500}$ for 6:4-0 and 6:4-1, $\frac{1800}{4500}$ for 6:4-2.
- $\frac{900}{5100}$ for 7:3-0, $\frac{2700}{5100}$ for 7:3-1 and 7:3-2.

As all papers using these datasets will face the same issue it will most likely not affect performance comparisons, but it will give a false sense of the potential improvement that could be made on these splits, as the maximum possible performance is less than 100%.

### A.3.2  UT-Zappos

UT-Zappos is a dataset of fine-grained types of shoes with material labels, where the material labels can not necessarily always be seen as a transformation that is grounded in vision. Many labels mostly

serve for humans to inform the material the shoes are made of, but are visually indistinguishable from other shoes with different material labels (cf. Figure 1). 8 out of 16 materials are some type of leather, and 3 materials are specifically designed to resemble other materials as much as possible (faux fur, faux leather, synthetic). Compound materials are only labeled with a single material, and not necessarily with the most prevalent material. Colors and patterns, combined with the low resolution of the images, often obscure crucial details and textures required to recognize certain materials.

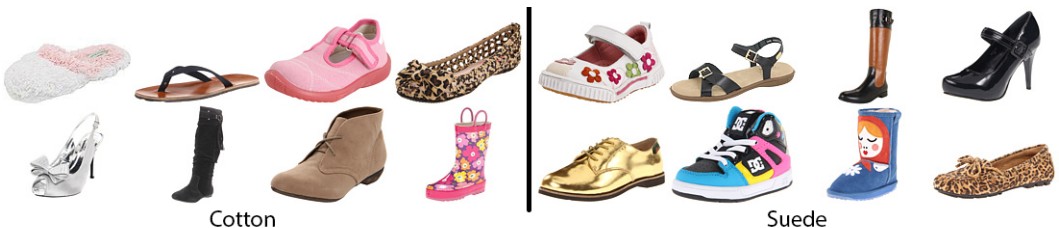

Cotton

Suede

Figure 1: **UT-Zappos**, examples of visually meaningless material labels.

### A.3.3 C-GQA

The C-GQA dataset [4] is a subset from the GQA dataset [5]. GQA is an inherently multi-attribute object detection dataset, but it has been filtered on the bounding boxes of large enough size where an object is only described by one attribute. However, due to the way object detection datasets are labeled, this still leaves many ambiguous images. Aside from the relatively high label noise, an object that is (partially) in front of another will be contained in the other object's bounding box. In one case, e.g., an image of a television may be labeled as a black television, while in another case it is labeled as the yellow wall it is hanging from (cf. Figure 2). Similarly, CZSL datasets for usually focus on one type of attribute of which there exists only one instance in the image, such as color or material, giving an unambiguous true answer to the question which attribute an image contains. GQA contains a wide variety of different attributes, even if an object has only been labeled with one, resulting in a high degree of ambiguity when only asking for a single attribute label.

### A.3.4 Side-note on Common CZSL Implementations

All CZSL papers with open source code that we have evaluated use imagenet crops (resize to $256 \times 256$, crop to $224 \times 224$). With the bounding boxes in GQA already being very tight, this crops out a considerable part of the subject of the image in the C-GQA dataset. The same happens, to a lesser extent, on UT-Zappos images, where the heel and tip of most shoes are cropped out, and to images in MIT-States.

### A.4 Error Analysis

### A.4.1 AO-Clevr

Aside from color and shape labels, AO-Clevr also provides size (small or large) and material (rubber or metal) labels. We look at the 4:6 split, where $47.0\%$ of all test samples are labeled 'small' (and $53.0\%$ 'large'), and $52.2\%$ are labeled 'rubber' (and $47.8\%$ 'metal'). For the default image size of $96 \times 96$, $82.1\%$ of all errors are made on test samples that are labeled 'small', indicating a significant limitation in recognizing small objects. Rubber vs. metal, or rather matte vs. shiny, does not offer significant difficulties, as $50.9\%$ of errors are made on samples labeled 'rubber', which is close to the expected error distribution.

To increase the performance on small objects, we look at two straightforward options: multi-scale architecture [6], and simply increasing the input image size. Yosinski et al. [7] have shown that neural networks learn more general, fine-grained features, such as corners or eyes, in their earlier layers, and global features representing entire objects in their final layers. A multi-scale architecture takes advantage of this fact by leveraging the outputs of multiple layers, instead of only the final layer. Figure 3 shows our multi-scale architecture. We repeat our prototype layer at 3 different output layers of the backbone, and sum the output logits for the final classification score. This approach brings the

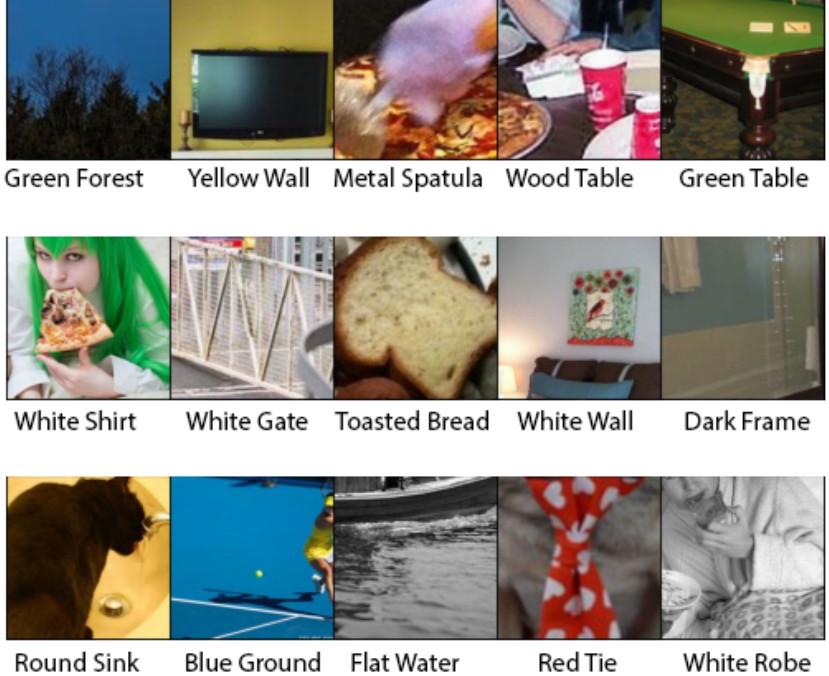

Figure 2: **C-GQA**, examples of images in the C-GQA dataset and their compositional labels, drawn uniformly at random.

error on 'small' samples down to $67.2\%$ (a $14.9\%$ absolute improvement), while also improving the final classification score.

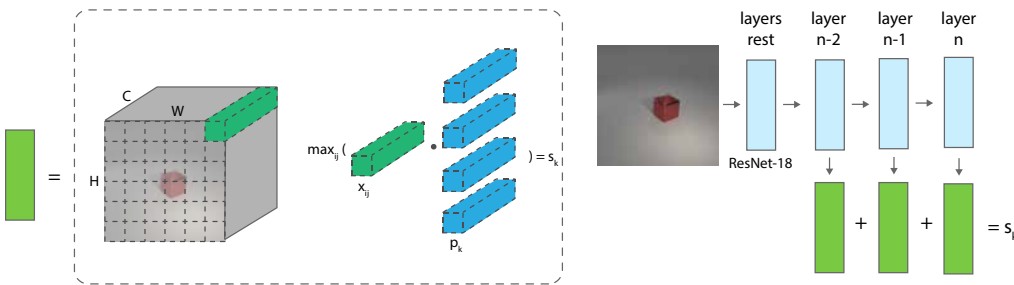

Figure 3: **Multi-Scale architecture**, prototype layers are placed at multiple scales in the backbone, their compatibility scores are combined for the final prediction.

Simply increasing the input image size to $224 \times 224$, however, leads to a much greater improvement, with a test sample error on 'small' objects of just $56.9\%$ (a $10.3\%$ absolute improvement over multi-scale). Both approaches significantly increase memory requirements, the multi-scale approach slightly less than increasing the image size. The multi-scale approach requires more computational resources, as the extra prototype layers require calculating the independence loss and spatial attention weights multiple times. Even though the final error of $56.9\%$ (as opposed to the expected $47.0\%$) still indicates some difficulty with recognizing small objects, utilising both approaches at the same time does not improve the final accuracy.

Finally, Figure 4 shows confusion matrices for the attribute and object predictions. We can see that almost all attribute errors misclassify a purple object as a red object. This can be explained by the

fact that purple and gray are the only colors that appear in just one compositional class in the training data, and purple is close to red in RGB values. For the objects, cubes and spheres are never confused, but cubes and spheres are sometimes misclassified as cylinders and vice versa. This can be explained by the fact that cylinders combine properties of spheres and cubes (both a flat and a rounded side). The error of confusing spheres for cylinders is most prevalent, as the training data contains twice as many cylinders and cubes as spheres. As the unseen:seen split increases we can see these kinds of errors exacerbate; blue and cyan start being confused, as well as red, brown, and yellow.

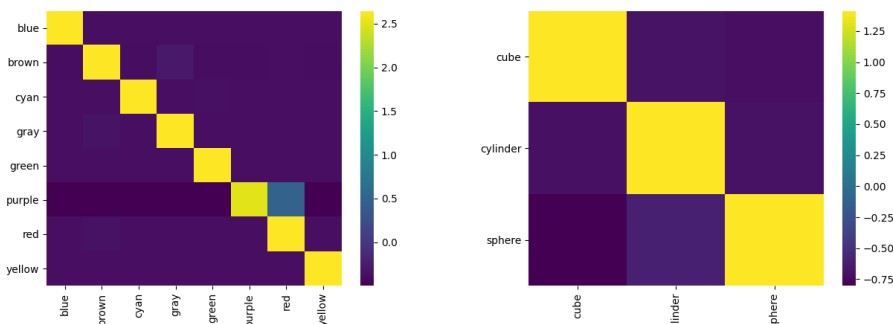

Figure 4: **AO-Clevr confusion matrices** for the 4:6 split.

### A.4.2 UT-Zappos

Figure 5 shows confusion matrices for the attribute and object predictions on UT-Zappos. For the object classifications most errors are fairly straightforward; knee-high boots, e.g., are confused with mid-calf boots, since the white background and cropping of the images often makes it impossible to infer scale. Boat shoes and slippers often have exactly the same shape as certain loafers.

The same can be seen for the attribute predictions, but as explained in Appendix A.3, the labels here are often visually indistinguishable. Faux leather, a material that has been explicitly designed to look as much like real leather as possible, is almost exclusively misclassified as real leather. Full-grain leather is often visually indistinguishable from regular leather. Suede and Nubuck are both a type of sanded leather, which are difficult to distinguish when colored and in low resolution images. Synthetic and Cotton shoes, again, attempt to emulate different material types (cf. Figure 1 for examples). Most of the errors are therefore mainly caused by attribute misclassifications; $59.7\%$ of the errors have the correct object label, $11.4\%$ have the correct attribute label, and the remaining $28.9\%$ misclassify both.

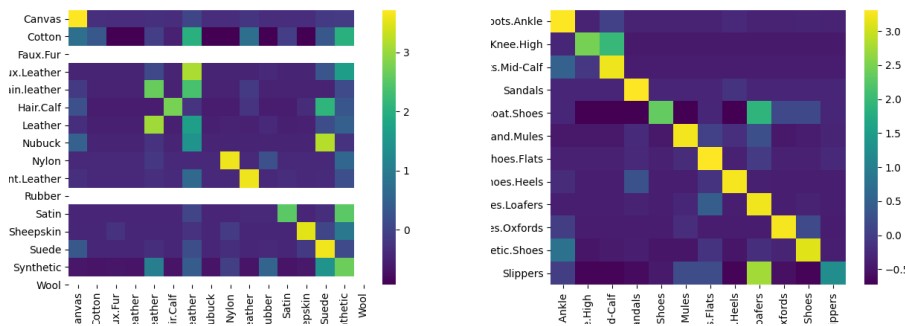

Figure 5: **UT-Zappos confusion matrices**, Faux.Fur, Rubber, and Wool are not part of the test set, and therefore displayed blank.

Table 1: **AO-Clevr Results**: The exact results as shown in the chart of the main paper.

| | ProtoProp (ours) | | | Causal [1] | | |
|---|---|---|---|---|---|---|
| U:S | Seen | Unseen | Harmonic | Seen | Unseen | Harmonic |
| 2:8 | $98.6 \pm 0.6$ | $99.3 \pm 0.4$ | $98.9 \pm 0.3$ | $89.7 \pm 1.9$ | $77.7 \pm 1.4$ | $83.2 \pm 1.2$ |
| 3:7 | $96.3 \pm 0.9$ | $81.7 \pm 0.1$ | $88.4 \pm 0.4$ | $80.9 \pm 3.6$ | $72.2 \pm 1.0$ | $75.7 \pm 2.3$ |
| 4:6 | $97.9 \pm 1.0$ | $95.5 \pm 0.9$ | $96.7 \pm 0.7$ | $84.1 \pm 1.8$ | $67.4 \pm 2.0$ | $74.7 \pm 1.7$ |
| 5:5 | $96.7 \pm 0.2$ | $63.1 \pm 0.4$ | $76.4 \pm 0.3$ | $83.8 \pm 0.8$ | $47.1 \pm 4.5$ | $59.8 \pm 3.9$ |
| 6:4 | $95.6 \pm 3.7$ | $38.6 \pm 4.7$ | $54.6 \pm 3.9$ | $86.1 \pm 2.9$ | $26.9 \pm 0.5$ | $40.9 \pm 1.0$ |
| 7:3 | $91.5 \pm 4.6$ | $39.3 \pm 1.6$ | $54.9 \pm 0.6$ | $69.3 \pm 6.1$ | $22.8 \pm 3.0$ | $33.7 \pm 4.2$ |

| | CGE [4] | | | TMN [8] | | |
|---|---|---|---|---|---|---|
| U:S | Seen | Unseen | Harmonic | Seen | Unseen | Harmonic |
| 2:8 | $96.7 \pm 2.0$ | $96.0 \pm 1.1$ | $96.4 \pm 1.5$ | $85.8 \pm 0.9$ | $79.7 \pm 4.4$ | $82.4 \pm 2.1$ |
| 3:7 | $98.2 \pm 0.7$ | $74.4 \pm 3.4$ | $84.6 \pm 2.5$ | $86.5 \pm 0.3$ | $62.2 \pm 4.9$ | $72.0 \pm 3.4$ |
| 4:6 | $95.5 \pm 1.1$ | $80.4 \pm 1.3$ | $87.3 \pm 0.7$ | $83.8 \pm 2.6$ | $68.1 \pm 4.1$ | $74.5 \pm 1.6$ |
| 5:5 | $94.1 \pm 2.5$ | $55.4 \pm 0.4$ | $69.7 \pm 0.9$ | $84.7 \pm 2.2$ | $38.0 \pm 3.0$ | $51.5 \pm 2.2$ |
| 6:4 | $95.4 \pm 0.1$ | $33.2 \pm 0.7$ | $49.3 \pm 0.7$ | $83.7 \pm 0.4$ | $18.1 \pm 2.9$ | $29.1 \pm 3.7$ |
| 7:3 | $77.9 \pm 0.1$ | $22.3 \pm 0.2$ | $34.7 \pm 0.3$ | $88.1 \pm 2.5$ | $5.8 \pm 0.9$ | $10.8 \pm 1.6$ |