# OpenReview forum: "Independent Prototype Propagation for Zero-Shot Compositionality"
_NeurIPS.cc/2021/Conference — NeurIPS 2021 Poster_

### Official Review · Reviewer_nHAS · 2021-07-06

**Rating:** 7
**Confidence:** 4

**Summary:**

The paper deals with the problem setup of compositional zero-shot recognition. Where a visual object is specified by two labels: an attribute name and an object name. Not all attribute-object combinations are available at training time, and at test time, a model may observe new (unseen) combinations.

The approach combines ideas from two recent publications: [1] Atzmon et al, 2020, and [2] Naeem et al, 2021: It uses the idea of encouraging independent prototypes for the attribute and object (Atzmon), and then it uses them as the node features for the graph-NN used by (Naeem), replacing than the word-embedding used in (Naeem).
In addition, it proposes a spatial attention mechanism over CNN feature maps for extracting the prototypes, rather than the average-pooling used by previous baselines.



**Limitations And Societal Impact:**

Limitations were adequately discussed


**Main Review:**


### Originality:
This work provides a novel combination of recent techniques (Atzmon, 2020) and (Naeem, 2021) - the idea is simple, and may be considered as immediate, but that's what I find as appealing in this approach.

### Quality:
The approach is technically sound, however there are issues with several key points:

* It will be beneficial to report the results of the model with the recent C-GQA benchmark (Naeem 2021).

* One claim uses a wrong terminology, but it is not a major issue because this is an architecture paper rather than a theoretical paper. Specifically, the paper claims that its loss function promotes *conditional*-independence of prototypes (as in Atzmon), where in fact, the loss term in eq (3) promotes *"full"* independence between prototypes. Promoting full independence may be problematic when used as a hard constraint (e.g. with a very large loss coefficient), because the attribute and object are not fully independent in the training data, but empirically it can work to some extent with a careful hyper-param tuning.

* Some claims are supported by the ablation study. However evidence is missing to several other claims. Specifically: (1) Superiority of Gaussian Kernel over Linear Kernel. (2) Superiority of a dot product similarity metric (3) superiority of simple linear combinations (4) benefit of cluster and separation loss.

* The reported metrics for Zappos are missing the Open Seen/Unseen metrics

* It will be more fair to cluster the compared approaches according to fine-tuned vs non fine-tuned.

* It will be beneficial to include an ablation study for the attention mechanism vs average pooling.

### Clarity:
The paper is mostly clear. Some sections  were hard to understand:

* Except for the linear activation, does the graph-NN architecture is identical to Naeem et al? Please discuss that in related work.
* For the HSIC kernel, how was the median distance between points calculated? If I understand correctly, this distance changes every time the model weights are updated. Therefore, it is not clear how to select this value if the distance keeps changing after every gradient update.
* Figure 2 and its caption is unclear.
* Paragraph in lines 147-154

### Significance:
Compositional generalization is a challenging and important problem in machine-learning models. The approach provides a simple combination of recent techniques using spatial attention, and provides promising results.



### Additional comment to authors:
* The claim "we do not use a fully connected layer" is not accurate: Each of the learned prototypes may be viewed as a vector of a matrix that parametrizes a fully-connected layer.


### Post rebuttal:
I thank the authors for their answers. Following the rebuttal I decided to raise my score to 7.


**Time Spent Reviewing:**

5

---

> ### Author Response · Authors · 2021-08-09
> **Thanks and response**
>
> > It will be beneficial to report the results of the model with the recent C-GQA benchmark (Naeem 2021).
>
> We have done some initial experiments on C-GQA. We have not been able to do much hyperparameter tuning yet, but our initial results are promising: currently a 15.1% harmonic mean vs. the state of the art of 14.5%.
>
> The dataset mostly fits in our original experimental setup, with an additional hyperparameter that was explored for the other datasets too, but found to be not necessary there. Specifically, we average-pool the $7 \times 7 \times C$ output of the convolutional backbone to $2 \times 2 \times C$. This is necessary because the images in C-GQA have been cropped with a very tight bounding box already, causing the local patches to be much more fine-grained than in the other datasets. To avoid overfitting we additionally needed to add a dropout layer, with a dropout of 0.3, to our graph layers.
>
> ---
>
> > One claim uses a wrong terminology, [...] Specifically, the paper claims that its loss function promotes conditional-independence [...] the loss term in eq (3) promotes "full" independence [...] Promoting full independence may be problematic when used as a hard constraint (e.g. with a very large loss coefficient), because the attribute and object are not fully independent in the training data, but empirically it can work to some extent with a careful hyper-param tuning.
>
> We thank the reviewer for pointing out this inconsistency, we will update our mentions of the independence loss to drop ‘conditional’. We indeed want the individual attribute and object representations to be completely independent, even though, e.g., if we know an animal is pink, that would be a strong indicator of a flamingo instead of an elephant. This way we can start with a blank slate, so to speak, when reintroducing these dependencies in the compositional prototypes. E.g., by propagating the features for the color “pink” to the object “flamingo”.
>
> We have found that, while we did carefully tune our loss coefficient, a large coefficient does not drastically degrade the accuracy of the final classifications. We will add evidence for this to our extended ablations in the appendix.
>
> ---
>
> > Some claims are supported by the ablation study. However evidence is missing to several other claims. Specifically: (1) Superiority of Gaussian Kernel over Linear Kernel. (2) Superiority of a dot product similarity metric (3) superiority of simple linear combinations (4) benefit of cluster and separation loss.
>
> 1) In our early experiments we have indeed considered both linear and gaussian kernels. The gaussian kernels always significantly outperformed the linear kernels. From that point on, we switched to the gaussian kernel exclusively. E.g., in one setting, a linear kernel reached a harmonic mean of 58.6, where the gaussian kernel reached a harmonic mean of 75.6, with all components and other parameters being equal. We will add ablations to table 2.
> 2) We found that the dot product reached its optimal point 10+ epochs earlier than cosine similarity and euclidean distance, and reached a much higher harmonic mean. We will  provide additional ablations in table 2.
> 3) Similar to (1), we switched once we found linear combinations performed better. We assume this is because the individual prototypes are already well-tuned to the visual domain, and don’t need complex nonlinear transformations such as with semantic embeddings. We will provide additional ablations in table 2.
> 4) The losses are directly adopted from Chen et al. [1], who provide evidence for their ability to improve prototypical representations (reference included in the paper).
>
> ---
>
> > The reported metrics for Zappos are missing the Open Seen/Unseen metrics
>
> For direct comparison, we reported the same metrics previous work have reported (AUC, Closed seen and unseen, and harmonic mean). We agree that the Open metrics have added value and we will add them to the paper.
>
> ---
>
> > It will be more fair to cluster the compared approaches according to fine-tuned vs non fine-tuned.
>
> We will make this distinction more clear in our results tables.
>
> ---
>
> >Figure 2 and its caption is unclear.
> >Paragraph in lines 147-154
>
> We will improve the writing of the mentioned paragraph and the explanation of the mentioned figure. Below follows a summary.
>
> In the prototype layer, the $1 \times 1 \times C$ input patch with the highest similarity score with a prototype is the localized representation of the attribute or object in a particular image. That specific patch is what we want to be independent from the other object or attribute representation, without affecting the other patches that do not encode an attribute or object.
> We could obtain the location of said patch via an argmax operation over the similarity scores, but that has a gradient of 0 almost everywhere. As such, we use attention pooling to collect the patches that encode the attribute or object as an approximation of the argmax operation.
> The pooling works by first calculating the prototype similarity score of each input patch. The softmax over these scores will result in, e.g., our target patch having a score of ~0.8, a neighbouring patch with a score of ~0.2, and a 0 for all other irrelevant patches. Our new approximated representation sums all patches with the aforementioned weights, which is differentiable.
>
> ---
>
> > It will be beneficial to include an ablation study for the attention mechanism vs average pooling.
>
> Average pooling would mix in all global information not related to the target attribute or object, which would then be included in the independence loss. We want to avoid that by design. We note that the attention mechanism is only used for collecting the local patch that encodes the attribute or object of interest, for use in the independence loss function.
>
> ---
>
> > Except for the linear activation, does the graph-NN architecture is identical to Naeem et al? Please discuss that in related work.
>
> Yes, both Naeem et al [2] and our model use a standard GCN layer, except for some minor differences, such as Naeem et al use a dropout of 50% on their graph inputs and L2-normalize the output, whereas we do not. We will make this explicit in our related work section.
>
> ---
>
> > For the HSIC kernel, how was the median distance between points calculated? If I understand correctly, this distance changes every time the model weights are updated.
>
> The median distance is recalculated for each batch, we will add this to the paper.
>
> ---
>
> > The claim "we do not use a fully connected layer" is not accurate: Each of the learned prototypes may be viewed as a vector of a matrix that parametrizes a fully-connected layer.
>
> We agree that the prototypes may be viewed as parameters of a fully connected layer.
> The claim in question (line 134-135) does not refer to the prototypes, but to an architectural difference to Chen et al. [1], who use an additional fully connected layer over the calculated prototype scores to calculate a final class score, whereas we do not.
> We will rewrite the paragraph to convey this properly.
>
> **References**
> [1] Chaofan Chen, Oscar Li, Daniel Tao, Alina Barnett, Cynthia Rudin, and Jonathan K Su, "This looks like that: Deep learning for interpretable image recognition." NeurIPS 2019
> [2] Muhammad Ferjad Naeem, Yongqin Xian, Federico Tombari, and Zeynep Akata. "Learning graph embeddings for compositional zero-shot learning." CVPR 2021

---

### Official Review · Reviewer_iHTC · 2021-07-15

**Rating:** 6
**Confidence:** 3

**Summary:**

The paper proposes ProtoProp model for compositional zero-shot learning, which propagates the learned object and attribute prototypes with independency constraints by a compositional graph of GNN. The compositional prototypes are then used for detecting unseen classes. Experiments are done on AO-Clever and UT-Zappos benchmarks.

**Limitations And Societal Impact:**

1) Using HSIC loss to maintain the attribute-object independence achieves improved embedding property but it seems a direct adaptation from Atzmon et with different kernels. But it has no experiment table to validate the improvement.

2) Missing detailed discussion or comparison to some related works on learning the prototype-based representations, such as [a, b]. It would be interesting to see the influence of prototypical representations obtained by the different methods.
[a] PANet: Few-Shot Image Semantic Segmentation with Prototype Alignment. ICCV, 2019.
[b] Prototypical Cross-Attention Networks for Multiple Object Tracking and Segmentation. Arxiv.

3) Missing detailed failure cases analysis, and qualitative visual results.

4) In Table 2, missing discussion on why finetuning is critical to visual node features but has smaller influence on the semantic node feature?

5) What's the influence of number GNN layers/design choices to the propagation and final performance.


**Main Review:**

1) The proposed ProtoProp model is validated on two zero-shot learning benchmarks with strong performance improvement.

2) The idea of compositional prototypes from attribute and object prototypes by GNN propagation is novel to me.

3) The paper writing is clear and well organized.


**Time Spent Reviewing:**

3

---

> ### Author Response · Authors · 2021-08-09
> **Thanks and response**
>
> > Using HSIC loss to maintain the attribute-object independence achieves improved embedding property but it seems a direct adaptation from Atzmon et al with different kernels. But it has no experiment table to validate the improvement.
>
> We will add this detailed validation of the improvement to our paper. The details are as follows: In our early experiments we have indeed considered both linear and gaussian kernels. The gaussian kernels always significantly outperformed the linear kernels. From that point on, we switched to the gaussian kernel exclusively. E.g., in one setting, a linear kernel reached a harmonic mean of 58.6, where the gaussian kernel reached a harmonic mean of 75.6, with all components and other parameters being equal. The new validation table will clarify this reasoning.
>
> ---
>
> > Missing detailed discussion or comparison to some related works on learning the prototype-based representations, such as [a, b]. It would be interesting to see the influence of prototypical representations obtained by the different methods.
>
> Thank you for bringing these papers to our attention. These are good examples of the versatility of prototypical representations in vision. We will add them to our related work section on prototype networks.
>
> We agree that it would be interesting to see the influence of different prototype representations on our overall model, but the focus of our paper is on how a given type of prototype representation can be improved by optimizing for independence and composing them through a graph model. To that end, we selected a suitable and common prototype representation, and consider a comparison of various prototype representations out of scope for our current work.
>
> ---
>
> > Missing detailed failure cases analysis, and qualitative visual results.
>
> We agree and performed a failure case analysis (error analysis), which we will provide as extended results in our appendix (it is lengthier than the allowed one page extension). We will also add qualitative results to the same appendix.
>
> What we noticed in this new analysis: most of the failure cases (errors) are fairly straightforward and explainable. The following is a short summary of the insights which will be added to the paper:
> * Most errors on AO-Clevr confuse cubes or spheres for cylinders, as cylinders have both flat and curved sides. Also, the model has more difficulty with objects that are labeled 'small'. The first error type is natural, because objects with more visual features are more likely to be confused for objects which share one (but not all) of those features. The second is also natural, due to less information being available in images of small objects.
> * On UT-Zappos, most errors are made on the “material” labels, mainly within the 8 different types of leather that often look alike even to humans. In a similar vein, knee-high and mid-calf boots are sometimes confused, as the white background and rescaling of the images often makes it impossible to infer scale, ie. that one is taller than the other.
>
> ---
>
> > In Table 2, missing discussion on why finetuning is critical to visual node features but has smaller influence on the semantic node feature?
>
> We have not performed a fine tuning ablation on the semantic node features. We will improve the clarity of Table 2 to convey this properly.
>
> Fine tuning for semantics is interesting and has been studied extensively in Naeem et al. [c] Given that our model is fully based on vision, without semantics, we wanted to understand both (a) the overall performance by a vision vs. a semantic model (by comparing our model to Naeem et al), and (b) the performance difference if semantic node features would be used instead of visual node features (ablation in Table 2).
>
> ---
>
> > What's the influence of number GNN layers/design choices to the propagation and final performance.
>
> We find that a 2-layer GNN is optimal. More layers results in oversmoothing, fewer layers results in worse regularization. These findings are in line with those of Naeem et al. [c]. We will add these findings to the appendix of the camera-ready version.
>
> **References**
> [a] PANet: Few-Shot Image Semantic Segmentation with Prototype Alignment. ICCV, 2019.
> [b] Prototypical Cross-Attention Networks for Multiple Object Tracking and Segmentation. Arxiv, 2021.
> [c] Muhammad Ferjad Naeem, Yongqin Xian, Federico Tombari, and Zeynep Akata. "Learning graph embeddings for compositional zero-shot learning." CVPR 2021

---

### Official Review · Reviewer_jpXL · 2021-07-15

**Rating:** 7
**Confidence:** 4

**Summary:**

The submission focuses on compositional zero-shot learning, in this context a type of zero-shot problem where instances are a pair of images and compositional labels (object/attribute information).
The authors start with the standard prototypical networks setup, and add auxiliary losses in order to learn object and attribute prototypes. A HSIC loss is added to encourage object-attribute independence (for better compositional generalization).
Finally, a graph propagation algorithm is applied to combine the prototypes into compositional prototypes.

**Ethical Concerns:**

I do not find any ethical issues with this paper.

**Limitations And Societal Impact:**

The authors have adequately addressed the limitations/potential negative societal impact. Indeed, they mention that there are no obvious issues, but that misuse can always happen, and that therefore caution should be exercised on a per-case basis.

**Main Review:**

Positive aspects:
- The paper is well-written, the ideas are clear.
- Empirical evaluation is satisfactory: there is an ablation study and a detailed discussion of the impact of the different components.

Limitations/to improve:
- I disagree with the author's claim that attribute datasets are either noisy or do not work in the single attribute setting. As an example, Caltech Birds (CUB) can be easily modified to account for a single attribute (e.g. color). More generally, one of my main issues is that the datasets considered are quite simple compared to the more complex "real-world" datasets more commonly seen in the classical zero-shot literature. Could the authors adapt their model and baselines to e.g. Caltech Birds or a related dataset and show that their approach indeed generalizes beyond toy datasets?
- The related works section is missing some works, notably [1] which is highly relevant to the discussion on compositional generalization. They propose a method, by no means the only one, to measure the compositionality of a representation. It would be interesting to see if there is indeed a link between a measure of compositionality (not necessarily theirs) and downstream performance of the representations in this submission. In a similar vein, I would mention [2] who take a more applied standpoint on compositional generalization.

After rebuttal: looking at the author response, I have raised my score to a 7. Cf. my response to their point for justification.

References
[1] Tristan Sylvain, Linda Petrini, and Devon Hjelm. "Locality and compositionality in zero-shot learning." ICLR 2020
[2] Wenjia Xu, Yongqin Xian, Jiuniu Wang, Bernt Schiele, Zeynep Akata,"Attribute Prototype Network for Zero-Shot Learning." NeurIPS 2020


**Time Spent Reviewing:**

4

---

> ### Author Response · Authors · 2021-08-09
> **Thanks and response**
>
> > I disagree with the author's claim that attribute datasets are either noisy or do not work in the single attribute setting. As an example, Caltech Birds (CUB) can be easily modified to account for a single attribute (e.g. color).
>
> The reviewer raises a good point, we will soften this claim in the text. Regarding CUB specifically, most birds consist of multiple different colors, which we believe would warrant multiple color labels per image, instead of the more restricted CZSL setting addressed in our paper, which would allow only a single color label per image. Extending to multiple labels is relevant and will be considered in our future work.
>
> ---
>
> > More generally, one of my main issues is that the datasets considered are quite simple compared to the more complex "real-world" datasets [...] Could the authors adapt their model and baselines [...] and show that their approach indeed generalizes beyond toy datasets?
>
> We have now included a new, real-world dataset: Naeem et al. [3] have released C-GQA very recently, a compositional zero shot adaptation of GQA visual question-answering dataset. This has over 9000 compositional classes, such as “brown horse” or “concrete walkway”.
>
> We have not been able to do much hyperparameter tuning yet, but our initial results are promising: currently a 15.1% harmonic mean vs. the state of the art of 14.5%.
>
> The dataset mostly fits in our original experimental setup, with an additional hyperparameter that was explored for the other datasets too, but found to be not necessary there. Specifically, we average-pool the $7 \times 7 \times C$ output of the convolutional backbone to $2 \times 2 \times C$. This is necessary because the images in C-GQA have been cropped with a very tight bounding box already, causing the local patches to be much more fine-grained than in the other datasets. To avoid overfitting we additionally needed to add a dropout layer, with a dropout of 0.3, to our graph layers.
>
> ---
>
> > The related works section is missing some works, notably [1] which is highly relevant to the discussion on compositional generalization. They propose a method, by no means the only one, to measure the compositionality of a representation. It would be interesting to see if there is indeed a link between a measure of compositionality (not necessarily theirs) and downstream performance of the representations in this submission. In a similar vein, I would mention [2] who take a more applied standpoint on compositional generalization.
>
> We thank the reviewer for their suggestion, [1] is highly relevant and interesting in terms of both locality and compositionality. We will add a discussion on this topic to the introduction of our paper.
>
> Regarding the measure of compositionality, we directly optimize for such a metric, which is explicitly linked to our downstream performance. Specifically, our compositional class score (cf. paragraph starting at line 187, and Figure 3) fits the definition for the TRE measure given by [1]. The TRE measure is defined as $\operatorname{TRE}(x, \mathbf{a} ; \eta)=\delta\left(f_{\eta}(x), f_{\eta}(D(x))\right)$, where $\delta$ is a distance function and $f_{\eta}(D(x))$ a composition of attribute representations. In the context of our compositional loss, the distance measure $\delta$ being our dot product similarity function, $f_{\eta}(x)$ the input image representation, and $f_{\eta}(D(x))$ our composition of the attribute and object representations.
>
> The work in [2] is certainly relevant too, regarding their use of local attribute prototypes. We had already cited [2] in relation to future work (line 294), and will add a mention in the related work section on prototypical networks.
>
> **References**
> [1] Tristan Sylvain, Linda Petrini, and Devon Hjelm. "Locality and compositionality in zero-shot learning." ICLR 2020
> [2] Wenjia Xu, Yongqin Xian, Jiuniu Wang, Bernt Schiele, Zeynep Akata, "Attribute Prototype Network for Zero-Shot Learning." NeurIPS 2020
> [3] Muhammad Ferjad Naeem, Yongqin Xian, Federico Tombari, and Zeynep Akata. "Learning graph embeddings for compositional zero-shot learning." CVPR 2021

---

> > ### Comment · Reviewer_jpXL · 2021-08-19
> > **Response to author comments**
> >
> > After reading the author's response I have decided to raise my score (to 7). The addition of another more complex dataset and the expansion of the discussion on compositionality justifies that in my opinion.

---

### Author Response · Authors · 2021-08-09
**Thanks and summary of responses**

We thank the reviewers for their time and constructive feedback. We are grateful for the kind words about the novelty of our method (R2, R3), strong and promising results (R2, R3), and clear writing and organization (R1, R2).

The reviewers raised some concerns, which we have addressed. Below we summarize future **key additions to the camera-ready version**:

* new results on a new, more complex and **real-world benchmark**: C-GQA
* new **extended ablations**, specifically, for the gaussian kernel, linear combinations in the GNN, and dot product similarity metric.
* a new **detailed error analysis and qualitative results** (in the appendix).
* an **extended related-work** section with the suggestions provided by the reviewers.
* improvements in the clarity of the attention pooling part of our methodology.

These improvements are described in the responses to each reviewer.

---

### Decision · Program_Chairs · 2021-09-27

**Decision:**

Accept (Poster)

**Comment:**

Compositional generalization is an important problem in machine learning. The problem setup is well motivated and the paper is well written. The proposed method is simple and effective.  It combines several existing techniques with innovations. All reviewers agree the acceptance.